# Perceived benefits of community-based TB preventive treatment in children in Uganda: "When she sees other children getting the same medication, she will feel not alone."

Elijah Ronald Kakande[1]*, Jason Johnson-Peretz[2], Rachel Abbott[2], Bob Ssekyanzi[1], Maxim Twinomujuni[1], Fred Atwiine[1], Milliam Korukiiko[1], Hellen Nakato Atuhaire[1], Joan Nangendo[3], Gloria Nattabi[1], Willington Ariho[1], Geoff Lavoy[1], Gabriel Chamie[2], Laura B. Balzer[4], Diane V. Havlir[2], Fred C. Semitala[1,5], Edwin Charlebois[6], Moses R. Kamya[1,5], Carina Marquez[2]

1 Infectious Diseases Research Collaboration, Kampala, Uganda, 2 Division of HIV, Infectious Diseases, and Global Medicine, University of California San Francisco, San Francisco, California, United States of America, 3 Implementation Science Program, School of Public Health, Makerere University College of Health Sciences, Kampala, Uganda, 4 Division of Biostatistics, University of California Berkeley, Berkeley, California, United States of America, 5 Department of Medicine, Makerere University College of Health Sciences, Kampala, Uganda, 6 Department of Medicine, Center for AIDS Prevention Studies, Division of Prevention Science, University of California, San Francisco, San Francisco, California

* rkakande@idrc-uganda.org

## Abstract

Tuberculosis preventive treatment (TPT) uptake among children at risk for TB remains low in sub-Saharan Africa. Community-based TPT delivery is effective at increasing uptake and completion in children compared to clinic-based models, but implementation research is needed to inform scale-up in real-world settings. In this qualitative study, we assessed community and health care provider perspectives on the anticipated benefits and barriers of a community-based TPT differentiated delivery model comprising three components: (1) initial screening and TPT initiation at the nearest public health facility; (2) community health worker (CHW)-led TB education with community-based medication delivery; and (3) CHW-facilitated delivery in a Community Adherence Group (CAG). From 5 September to 12 October 2023, we conducted in-depth semi-structured interviews (N = 20) with a purposively selected sample of six public health care providers, four CHWs, and ten caregivers of children with latent TB infection. A six-person multi-regional research team translated and coded transcripts. Framework analysis was used to identify perceived benefits and barriers. Participants identified five main benefits: (1) comfort receiving care in the community due to familiarity with differentiated HIV care models and trust in CHWs; (2) peer support in CAGs promoting adherence and reducing stigma; (3) reduced transport costs; (4) improved efficiency through reduced waiting times and provider workload; and (5) enhanced CHW capacity to provide TB prevention services and

**Data availability statement:** Data availability statement: Sharing unredacted transcripts may allow readers to identify particular people, thus ethically violating our confidentiality agreements with our participants. Following SEARCH Scientific Review Committee approval of a concept sheet that includes a summary of proposed analyses, secure curation of data, the intended use of said data, and a signed data access agreement, we can provide de-identified study data and a data dictionary via a secure online repository. Please direct further inquiries to the SEARCH Scientific Review Committee via searchprojectinfo@gmail.com.

**Funding:** This research work was supported by the U.S. National Institute of Health under award number R01AI151209. Elijah Kakande was funded under the Makerere University Implementation Science (MakImS) mentorship program under award number NIH/FIC D43TW010037. The funders had no role in study design, data collection and analysis, decision to publish, or preparation of the manuscript.

**Competing interests:** The authors have declared that no competing interests exist.

increase community awareness. Perceived barriers included low community knowledge, need for a consistent drug supply, stigma, and inadequate CHW training. Caregivers, healthcare workers, and CHWs identified peer support, trust in CHWs, reduced transport costs, and increased efficiency as key benefits. Implementation efforts should address these barriers to optimize delivery in rural East Africa and similar settings. Integration of CAGs into community-based TPT models warrants further study.

## Background

Tuberculosis preventive therapy (TPT) is an important strategy to end childhood tuberculosis (TB) and reduce the risk of active TB among children under five years who have an elevated risk of progression to active disease and acquiring severe forms of the disease once infected [1–5]. However, uptake among children remains low, with 42% of 1.6 million children under five years receiving the medication in 2023 [1]. In Uganda, which is one of 30 high-TB burden countries, the major entry point for TPT in children is TB household contact tracing, conducted by health care providers and clinic-based community health workers (CHWs) [1,6]. While this strategy is effective in identifying children at the highest risk for TB, the number of children identified and initiated on TPT remains suboptimal [1]. Barriers to clinic-based TPT delivery include a lack of belief in efficacy and need for TPT, as well as practical barriers, including transport, long waiting hours at the clinic, and time constraints [7].

Community adherence groups (CAGs) focusing on children where TPT is delivered within the group by a CHW with healthcare provider support can overcome practical barriers of clinic-based TPT (long distances to the facility, long waiting times at the clinic, and frequent monitoring visits) and foster trust in services compared to clinic-based care [8]. A recent trial conducted in Cameroon and Uganda, where CHWs conducted household-based TB screening and TPT adherence monitoring with healthcare provider TPT initiation for children within the household, increased contact tracing coverage and TPT completion compared to clinic-based TPT delivery [9]. The expansion of community-based delivery to include CHW-led health education and TPT delivery in a CAG could have additional benefits to children and further increase TPT uptake, adherence, and completion. While CAGs have been mostly explored in the differentiated care of adults with HIV [10–14], they could be leveraged for CHW-led health education, TPT delivery, and peer support in children. There remain key scientific knowledge gaps about how best to implement community-based TPT delivery for children, and real-world implementation studies are needed to inform these efforts. Formative studies to inform the design of such interventions are critical. We conducted a qualitative study to explore the barriers and facilitators of a community-based TPT model, that included a CHW-facilitated community-adherence group for children via CHWs in rural Southwestern Uganda.

## Methods

### TB screening in the Socio-spatial Networks for TB infection in children and youth in rural Uganda (SONET) study

This study was embedded within the Socio-spatial Networks for TB infection among children and youth in rural Uganda (SONET), an epidemiological study that aimed to ascertain when and where children and youth acquire TB infection. The study screened all children aged 1–17 living in households in 25 villages in Ndeija subcounty in south-western Uganda (N = 5879) for latent TB infection using QuantiFERON Gold plus (QFT), a blood-based assay [15]. Children diagnosed with latent TB infection were referred to the nearest public health facility able to provide TB treatment and prevention services. Participants who qualified for TB preventive treatment would receive it at the health facility. After 1 year of follow-up, repeat TB infection status was ascertained in 66% of children.

### Qualitative study design

The goal of this qualitative study was to explore perceived benefits and barriers to community-based delivery of TB preventive treatment to children with latent TB infection from the perspectives of CHWs, parents/caregivers of children in the study, and health care providers in villages participating in the SONET study. Between 15th September and 12th October 2023, a trained team of qualitative researchers administered in-depth, semi-structured interviews to a purposive sample of; 1) Caregivers of children (n = 10) participating in the SONET study who also had a diagnosis of latent TB infection after 1 year of follow-up or who were living with HIV (these groups were purposefully selected because they would be the intended beneficiaries of TPT in CAGs); 2) CHWs (n = 4) who deliver services to households in participating villages, and health care providers (n = 6) including doctors, clinical officers, and nurses involved in the management and delivery of TB services at the public health facility within the study catchment area. Thematic saturation was assessed at the level of the overall dataset rather than within individual participant subgroups. Subgroup sample sizes were not intended to achieve independent saturation; instead, input from health care providers, community health workers, and caregivers informed the identification and refinement of the core themes.

Interview guides were tailored to each specific participant category (S1 Text - S3 Text). The guides were designed to collect information about participant demographics, knowledge about TPT, perceived advantages and disadvantages of differentiated service delivery models, and opinions about the integration of community-based TPT into current healthcare systems. The hypothetical differentiated service delivery model that we discussed with participants comprised three core components: 1) initial screening and initiation of TPT at the nearest public health facility, 2) Community Health Worker (CHW) led TB education in a CAG (where child TB contacts and their caregivers from households overseen by a CHW gather every 2–3 months at a community venue in their village for TB prevention services) and 3) CHW facilitated delivery of TPT in a CAG. Guides were translated into English and back translated into the local language (Runyankole) for accuracy. A team of two Ugandan researchers, [FA, MK] trained to conduct qualitative interviews and fluent in both English and Runyankole, interviewed participants. Interviews lasted 45–60 minutes and were conducted in the language that respondents were most comfortable with.

### Data analysis

After transcribing the interviews into English, a six-person multi-regional team [EK, CM, JJP, RA, BS, TM], including Ugandan researchers, reviewed and coded all transcripts in Dedoose using a codebook derived from *a priori* research questions (S1 Data). Using a framework analysis approach, we then identified key themes participants raised under the larger headings of perceived benefits, facilitators, barriers, and recommendations for successful delivery of community-based TPT.

### Ethical approval

The study was approved by the School of Medicine Research Ethics Committee of Makerere University College of Health Sciences (221–166), the Uganda National Council for Science and Technology (HS1844ES), and the Human Research

Protection Program and Institutional Review Board of the University of California San Francisco (20–32231). All participants provided written informed consent prior to participation.

### Inclusivity in global research

The authors have taken several steps to ensure the inclusivity of this study. This research was developed in equal partnership with investigators in Uganda, who were involved in study design, data collection, and the interpretation of findings. To ensure the research met local priorities and cultural norms, we engaged with community stakeholders through a representative Community Advisory Board throughout the process. Additional information regarding the ethical, cultural, and scientific considerations specific to inclusivity in global research is included in the Supporting Information (S1 Checklist).

## Results

Below, we present caregiver ("patient"), Community health worker, and health care provider perspectives towards community-based TPT. The health care providers included 3 health managers (District Health Officer, District TB focal person, and Health facility In-charge) [Table 1]. We first present the five general themes participants identified as benefits of community-based TPT delivery: peer support, trust in CHWs, reduced transport costs, reduced burden at clinic, and enhanced efficiency. We then present four themes on barriers to the successful implementation of a community-based TPT strategy: need for constant drug supply, need for training and supervision for CHWs, low community knowledge, and stigma and confidentiality concerns.

### Benefits

**Peer support in community TPT groups can promote adherence and reduce stigma.** Participants felt that children would be encouraged to take their medication and feel less stigmatized, seeing that they are among other children in the community group, all receiving TPT.

*"When she sees other children in the same group also getting the same medication, she will feel she is not alone, and that will reduce the stigma of thinking that she is suffering from latent TB and won't get worried about that condition because she knows she is not alone."* -Female caregiver of a 6-year-old female child

**Comfort receiving care in the community and trust in CHWs.** A key component of the proposed intervention was delivery of TPT in CAGs, aided by a CHW. This approach was viewed as more comfortable since CHWs are part of the community and are likely to be trusted more by the community compared to health care providers, as one health care provider stated:

*"I recommend it 150% because even with HIV in management of people who are not suppressing, who have adherence issues, with the 2022 guidelines, we have shifted to the community, those significant people who are comfortable*

**Table 1. Description of study participants.**

| Cadre | Male | Female | N=20 |
|---|---|---|---|
| Health care providers (including 3 health managers) | 1 | 5 | 6 |
| Community Health Workers | 2 | 2 | 4 |
| Caregivers of children exposed to TB | 2 | 8 | 10 |
| Children whose caregivers were interviewed<br>Diagnosed with Latent TB<br>Children living with HIV | 3<br>2 | 4<br>1 | 7<br>3 |

*getting services from people they know, serving a client at their comfort in the community for me, it is the best approach.*" Female health care worker

"*It is okay because she [CHW] is near us and she lives in our community. If she is first trained in what to do, I don't see any problem with her doing this job.*"

- Female Caregiver of 6-year-old female child

**Receiving TPT in the community will reduce transport costs.** One of the key barriers to facility-based TPT is the required investment in transport costs to sustain follow-up visits [7]. Respondents perceived that community-based TPT delivery would directly tackle this barrier, as one caregiver remarked: "*What I have liked about it is we shall be saved from spending on transport because one person will be going there and come and give to us.*" - Female caregiver of a 4-year-old female child

**Community based TPT delivery reduces patient lines, waiting times, and provider workload at the health facility.** For community members, facility-based healthcare affects daily activities and leads to loss of productive time due to travel and waiting in long queues to access services. Community-based TPT delivery addressed these barriers in addition to reducing health care provider workload at the facility.

"*This is a very good thing because we have a lot we do in a day and on a day's program if there is going to the facility to get the medicine, it means those daily activities get affected…. and there are long queues from the health facility, but if the medicine is got from our community the CHW will communicate to us that on this day and at this time you come for the medicine for your children, or the children will come for the medicine.*" -Female caregiver of a 6-year-old male child

"*It will [be convenient] because in one way or the other this approach will be helping them, since VHTs will be reducing their work load at the health facilities, we have been receiving many children for these refills but if they can now see that, there is now someone who can take for them the drugs to these children, they would have relieved them on their work load at the facility, and they will even have the good outcomes for those children who are initiated on the TPT getting their refills from the community.*" – Female health care worker

**CHWs enhance the efficiency of the model to provide TB prevention services.** One of the key advantages of working with CHWs is person-centered care. TPT delivery can be conducted at a time that is convenient for clients, increasing efficiency. Participants perceived that it would be easy for CHWs to adjust their schedules to match clients' availability, as one health worker commented: "*The CHWs is flexible; she can even schedule them in the afternoon to be the time to come for their refills when they are done with their work in the morning hours, and which may not be the case at the health facility.*" Female health worker

"*When the medicine is in the community, even if the child is not around, say maybe he has gone to school, the mother can pick it for him. And another thing with these older children, they may not need to go with their parents where the community group meets to pick the medicine, they can go alone and get the medicine by themselves.*" – Female caregiver of a 5-year-old male child

### Barriers and drawbacks

Health care providers, CHWs, and caregivers shared anticipated barriers to this community-based approach, including system-level factors such as drug supply, provider-level requirements, such as training, and community-level drawbacks, such as low community knowledge and stigma.

**TPT delivery is dependent on a constant drug supply.** Health care providers recognized that community-based TPT was a good approach but would require a constant supply of drugs to be sustainable, as stated by one of the health workers: *"We should ensure that the drugs are available all the time for the CHW to pick and take to their communities to refill these children."* Female health care provider

**CHWs lack sufficient training, supervision, and logistics to deliver TPT.** CHWs identified the need for training to understand TB prevention and how best to deliver TPT in the community.

*"They should also first train us on what to do and how to do it, and when they train us well, we can really do this job, because if we don't get trained, we may not be able to perform our duties well."* Female Community Health Worker

Additional barriers identified by CHWs included the lack of logistical supplies, such as gumboots, umbrellas, and back-packs, to support them in executing their daily duties, and bicycles to aid movement to and from the health facility and within the community.

*"Also, during the rainy season, our village is slippery, and we may need gumboots and an umbrella to deliver well. For the issue of transport, it's not all about money for transport; they can also decide to give us bicycles, and that can help us ride to the health facility to pick up the drugs and come back."* Male Community Health Worker

**Low community knowledge about TPT.** The community's lack of knowledge about TB in general was perceived to be an important barrier to the successful implementation of community-based TPT. Low community awareness about TB is likely to affect the uptake of TPT by community members who may fear initiating preventive treatment, confusing it with TB disease treatment.

*"Yes, for some people they will think that the children are sick of TB and that it is the reason why they are taking the medication, and this will also somehow affect their parents psychologically, but when their parents get trained and sensitized first before initiating their children on the medicine, it will help them to catch up with such reactions from the community."* Male Community health worker

*"We need to first get trained about the approach, you see even when parents get trained it will ease our work because you mentioned that the children will be from 0-12 years in those children below 5 years may not really understand, but those of 12 for them they understand, so that is why I am suggesting that even parents of these children should be trained about this approach because that will ease the work of a CHW, I don't see any problem if the whole community get sensitized by the way, because even those children who don't have latent TB today may develop it tomorrow, so the whole village can be mobilized in one place and get sensitized about latent TB as this may even help to reduce the stigma I earlier mentioned about of course not all will embrace it but once the majority gets sensitized they will embrace it."* Male Community Health Worker

**Stigma and confidentiality.** Caregivers felt that delivery of TPT within a CAG via CHWs may lead to other community members finding out that their children have latent TB, raising concerns about CHW confidentiality and fear of being labelled as having active TB. Healthcare providers shared a similar view, raising concerns about CHW ability to maintain confidentiality.

*"It's like when you have tested HIV positive and you want start on HIV medication and they tell you that a CHW will be getting your medicine and delivering that medicine to you at your home, I rather come and get it myself to keep the disease that I am suffering from between me and the health workers, but not the CHW whose child may not be having*

*latent TB and then gets to know that my child has latent TB and she starts telling everyone in the community."* – Female caregiver of a 5-year-old female child

*Another challenge that we have seen in other programs, that we think may happen could be stigma, some of these CHWs because they are in the community, some clients may not be so comfortable with the CHW being involved in their health affairs, that is also something that we have seen, you tell a client about a service through a CHW in their area and they tell you no please! that one has a lot of rumor monger, so that is also a challenge that I fore see, but if these CHWs are taken through the right training, confidentiality trainings then they will be able to work well,"* – Female Healthcare Worker

## Discussion

In this qualitative study, healthcare providers, CHWs, and caregivers of children with TB infection felt that if implemented, a community-based delivery model with a CAG for TPT for children facilitated by a CHW would be beneficial for providing peer support for adherence. Key stakeholders universally highlighted that a community-based model would address barriers to accessing clinic-based care, including transport and productive time lost in waiting lines. They all felt that the model would be feasible and highlighted a unified desire to help the community. Though all agreed on the positive impact on structural barriers such as transport, caregivers noted some drawbacks to this model, including stigma and concerns about confidentiality. Some caregivers preferred home-based care over CAGs, in part because of concerns of confidentiality, suggesting a need for choice between these two forms of community-based TPT delivery. Respondents also felt that without increased community knowledge, CHW training, and logistical support, it would be challenging for this model to succeed. Delivery of TPT to children in CAGs could supplement clinic-based care [16] and community-based contact tracing strategies [17] to improve TB screening, TPT delivery, and monitoring for children.

Both health care providers and most community members perceived that delivery of TPT for children within a CAG would create a sense of togetherness and peer support. The concept of drug delivery within a peer support group has been extensively explored within the context of HIV care, where groups of up to 30 people, meeting routinely (every 2–3 months), access their treatment from the health facility via one representative [10–14,18,19]. These differentiated care models reinforce adherence and foster peer support at the community level. However, these have mostly been implemented for adults and not children, whose individual needs may not be catered to in an adult community group. Our study suggests that CAGs of TPT may be beneficial to children as well. To our knowledge, we are the first to describe the potential benefits of a CAG for TB preventive treatment in children.

Child-to-child peer support groups may help children to understand concepts about TB prevention and learn by copying their peers [20], facilitating adherence and completion of TB preventive medication. Additionally, while the integration of TPT delivery into differentiated HIV care caters to the needs of people with HIV, it may inhibit access to TPT for people without HIV for whom TPT is indicated. To our knowledge, this is the first study to explore a CAG for TPT for children, and our data suggest it may be acceptable and warrants additional study.

CHWs were perceived by caregivers and healthcare providers as a trustworthy source of healthcare information, who would deliver TPT within the community. CHWs are part of the health infrastructure in many low-income settings, including Uganda [21,22], bridging the gap between health facilities and the household. Their roles in TB prevention and care include health education, TB symptom screening, sputum collection, and adherence support through directly observed therapy [23]. In a cluster randomized trial conducted in Uganda and Cameroon, CHWs participated in TB screening and follow-up of people taking TPT medication to great effect, leading to increased contact tracing coverage and TPT completion compared to clinic-based TPT delivery [9]. In our proposed differentiated model, CHW roles would be expanded to include provision of health education and delivery of TPT to children in CAGs. CHWs carry the benefits of being part of the communities they serve, having personal relationships and the trust of community members, and being able to implement

flexible scheduling of appointments, unlike clinic-based care providers [24]. It was perceived that having CHWs as part of this delivery model would improve its efficiency and expand delivery of TPT. This is consistent with what others have reported about CHW-based TB service delivery strategies. For instance, in Ethiopia and Lesotho, CHWs conducting TB active case finding and follow-up through household visits improved TB contact screening, and TPT initiation and completion for both adults and children [25,26]. Both studies leveraged CHWs to deliver home-based TPT services compared to clinic-based TPT. In our proposed approach, TPT would be delivered to children in a CAG, potentially increasing benefits to children to improve adherence and completion of TPT.

However, while CHWs were perceived as trustworthy members of the community, caregivers and health care providers had reservations about stigma and CHWs maintaining confidentiality, with caregivers fearing that their health information could be shared with other community members, a concern that is commonly highlighted in community settings [27–30]. CHW training on proper handling of client information and improved CHW supervision could help address this concern [27,31]. In addition, it is important to acknowledge that not all people will prefer community-based services provided via CHWs. Provision of choice of either clinic-based, home-based or community group-based delivery could provide an alternative to people who are uncomfortable with CHW services or participating in groups. We have learned from our work elsewhere, offering Dynamic HIV prevention choice, that when offered options, people will make different choices which will change over time based on preferences and changing circumstances [32–36]. Choice-based models are person-centered and respond to client needs and preferences. In addition, health education embedded within the CAG is one creative way to increase TB knowledge and address negative attitudes and beliefs towards TB prevention at the community level.

Health care providers and CHWs cited training of CHWs as an important component of this community-based model. Health care providers perceived that CHWs lacked sufficient training to adequately screen children for TB and monitor them for side effects of TPT in CAGs and preferred to have community groups staffed by health care workers instead. This is a plausible alternative that has been explored in other low-income countries, such as Zimbabwe [12,37]. However, understaffed health facilities may not be able to sustain healthcare worker-facilitated provision of TPT in community groups, reducing efficiency [38]. Other system-level barriers to TPT initiation and completion for child contacts have been well documented, including human resource deficits, frequent stockouts of drug supplies, and funding shortages [39]. While the community CAGs are responsive to human resource deficits, they could be negatively impacted by drug stockouts and low funding. These system-level barriers, however, are not limited to CAGs and may impact other community-based approaches as well as clinic-based delivery, highlighting the need to strengthen the supply for TB commodities for successful implementation and scale-up of TPT for children.

Amuge and colleagues identified individual-level barriers to TPT initiation among children and adolescents, including carers lacking time to take children to the health facility, perception that children shouldn't swallow drugs when they are not ill, and stigma [7]. However, this study was conducted among people with HIV in a clinic-based model, and while some barriers identified (such as stigma) may be similar in both clinic and community settings, others may not be generalizable to different contexts (such as the community setting) and populations (such as children without HIV). Our study expands our understanding of barriers to TPT implementation to children beyond the health facility.

While we specifically assessed a community-based model with CAGs, our findings can also inform implementation of community-based models that include home delivery. CHWs emphasized the importance, in addition to training, of providing tools such as bicycles, gumboots, umbrellas, and enhancing communication with health care facilities, to enable delivery of TPT services to the community. Despite an increasing interest in expanding CHW roles to the provision of TB prevention and care [34,35], CHWs are currently volunteers, with limited government support [36,37], and need to be equipped and facilitated to work. One of the solutions proposed by the government is the creation of community health extension workers (CHEWs) [40], a new cadre of community health workers with a higher level of education, more extensive training, and some remuneration, who will provide support and supervision to groups of community health workers at

 

the village level. The link between CHWs and the health system can be strengthened further using creative avenues such as telehealth, with the clinician on the phone during follow-up visits to ensure sufficient monitoring for side effects and appropriate referral to a clinician for management, an approach that has been explored by others extending hypertension services to the community with the help of CHWs [38].

Our study has some limitations. First, the study explored the feasibility of a hypothetical community-based TB program, seeking to understand barriers to TPT implementation using CAGs for children prior to actual implementation. While this is highly informative and could aid the development of a sustainable and highly effective program, it may not reflect actual barriers that people will face during the implementation of such a program. Second, we did not explore the perspectives of managers at the level of the Ministry of Health, which would have expanded our understanding of the feasibility and sustainability of CAGs for TB prevention in children. However, we included the perspectives of the District Health Officer and District TB focal person who oversee the implementation of TB guidelines provided by the Ministry of Health in their district to provide insights into the sustainability of this approach. Finally, as a pilot study, we kept our sample population small. We therefore do not claim data saturation for each subgroup within the sample (e.g. CHWs). However, we did strive to include participants from each of the main stakeholder groups: providers, caregivers, and CHWs."

## Conclusions

Our data highlight implementation recommendation for community-based TPT models, and highlights acceptability of integrating CAGs. Caregivers perceived community-based TPT delivery with a CAG for children with the help of CHWs to be beneficial, address structural barriers to TPT access, and foster adherence due to peer support within groups. This approach may increase TPT uptake and completion among child contacts of TB cases in low-income, high-burden settings by complementing household contact tracing efforts. However, for this approach to be successful, important barriers to implementation, such as community stigma and lack of CHW training, could be addressed through community sensitization about TB, CHW training, allowing for choice between community and facility-based TPT services, and use of telehealth to strengthen the link between CHWs and providers at the clinic. Additionally, TPT delivery for children in a community setting will require a person-centered approach that meets individual preferences and needs.

## Supporting information

**S1 Text. Patient Qualitative interview guide - Children.**
(DOCX)

**S2 Text. Key Informant guide - Community Health Workers.**
(DOCX)

**S3 Text. Key Informant Guide - District managers and Health care providers.**
(DOCX)

**S1 Data. Qualitative data codebook.**
(PDF)

**S1 Checklist. Inclusivity in global health research.**
(DOCX)

## Author contributions

**Conceptualization:** Elijah Ronald Kakande, Jason Johnson-Peretz, Fred C. Semitala, Edwin Charlebois, Moses R. Kamya, Carina Marquez.

**Data curation:** Fred Atwiine, Milliam Korukiiko, Carina Marquez.

**Formal analysis:** Elijah Ronald Kakande, Jason Johnson-Peretz, Rachel Abbott, Bob Ssekyanzi, Maxim Twinomujuni, Carina Marquez.

**Funding acquisition:** Moses R. Kamya, Carina Marquez.

**Investigation:** Elijah Ronald Kakande.

**Methodology:** Elijah Ronald Kakande, Jason Johnson-Peretz, Moses R. Kamya, Carina Marquez.

**Project administration:** Elijah Ronald Kakande.

**Supervision:** Elijah Ronald Kakande.

**Writing – original draft:** Elijah Ronald Kakande.

**Writing – review & editing:** Elijah Ronald Kakande, Jason Johnson-Peretz, Rachel Abbott, Bob Ssekyanzi, Maxim Twinomujuni, Fred Atwiine, Milliam Korukiiko, Hellen Nakato Atuhaire, Joan Nangendo, Gloria Nattabi, Willington Ariho, Geoff Lavoy, Gabriel Chamie, Laura B. Balzer, Diane V. Havlir, Fred C. Semitala, Edwin Charlebois, Moses R. Kamya, Carina Marquez.

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
