## [Decision Letter · Decision Letter 0]

30 Dec 2025

PGPH-D-25-02798

Perceived benefits of community-based TB preventive treatment in children in Uganda: “When she sees other children getting the same medication, she will feel not alone.”

Dear Dr. Kakande,

Thank you for submitting your manuscript to PLOS Global Public Health. After careful consideration, we feel that it has merit but does not fully meet PLOS Global Public Health’s publication criteria as it currently stands. Therefore, we invite you to submit a revised version of the manuscript that addresses the points raised during the review process.

We look forward to receiving your revised manuscript.

Kind regards,

Abraham D. Flaxman, Ph.D.

Academic Editor

Journal Requirements:

Additional Editor Comments (if provided):

Reviewers' comments:

Reviewer's Responses to Questions

**Comments to the Author**

1. Does this manuscript meet PLOS Global Public Health’s publication criteria? Is the manuscript technically sound, and do the data support the conclusions? The manuscript must describe methodologically and ethically rigorous research with conclusions that are appropriately drawn based on the data presented.? Is the manuscript technically sound, and do the data support the conclusions? The manuscript must describe methodologically and ethically rigorous research with conclusions that are appropriately drawn based on the data presented.

Reviewer #1: Partly

Reviewer #2: Partly

2. Has the statistical analysis been performed appropriately and rigorously?

Reviewer #1: N/A

Reviewer #2: N/A

3. Have the authors made all data underlying the findings in their manuscript fully available (please refer to the Data Availability Statement at the start of the manuscript PDF file)?

The PLOS Data policy requires authors to make all data underlying the findings described in their manuscript fully available without restriction, with rare exception. The data should be provided as part of the manuscript or its supporting information, or deposited to a public repository. For example, in addition to summary statistics, the data points behind means, medians and variance measures should be available. If there are restrictions on publicly sharing data—e.g. participant privacy or use of data from a third party—those must be specified.requires authors to make all data underlying the findings described in their manuscript fully available without restriction, with rare exception. The data should be provided as part of the manuscript or its supporting information, or deposited to a public repository. For example, in addition to summary statistics, the data points behind means, medians and variance measures should be available. If there are restrictions on publicly sharing data—e.g. participant privacy or use of data from a third party—those must be specified.

Reviewer #1: No

Reviewer #2: No

4. Is the manuscript presented in an intelligible fashion and written in standard English?

Reviewer #1: Yes

Reviewer #2: Yes

Reviewer #1: With interest I have read this well-written manuscript about an important topic

1. Line 179 – Perspective in quotation. The quoted text uses a female pronoun (“she”), while the description indicates it refers to a male child. This creates ambiguity about whose perspective is represented—the son, daughter, or mother. Please clarify.

2. A limitation of the study is that program manager perspectives were not included. Incorporating these could provide valuable insights into the sustainability of this community-based intervention. Any discussion on sustainability would strengthen the manuscript.

3. Methods: The methods section needs more detail on the sampling approach. The authors mention a purposive sample, but the criteria for selection are not specified. Please elaborate on the basis for participant inclusion.

4. Demographic table. Include a table summarizing demographic characteristics of the interviewees. For CHWs and HCWs, it would be particularly useful to report their training background and tenure in their current roles.

5. appears to be an error with heading “1.2.3,” which is used twice. Please correct this duplication.

6. Sample size and data saturation. The sample size is small, and the manuscript does not address data saturation. This raises questions about the representativeness of the findings. It seems unlikely that saturation was achieved, especially with only four CHWs interviewed. Please discuss this limitation.

7. Line 347 – Missing reference. Add a reference for the statement on line 347, including the trial’s results and why they are relevant in this context.

8. Line 402 – Government proposal. Provide a reference to support the statement that the government is proposing CHEWs.

9. Discussion: The limitations of this study should be explicitly discussed in the discussion section.

10. Conclusion: In the conclusion, emphasize the importance of a tailored approach that meets individual preferences.

Reviewer #2: Review of Kakande et al.

This qualitative study reports on findings from in-depth interviews of caregivers and health care workers about a proposed community-based delivery model for TPT that includes adherence groups for support. The stakeholder perspectives are valuable, but the manuscript is unclear about how the proposed model of TPT delivery that was described to participants may have differed from the model they experienced.

Major

Introduction

1. Lines 96-99 indicate that “community adherence groups focusing on children where TPT is delivered within the group by a CHW with healthcare provider support can overcome practical barriers of clinic-based TPT…”. What is the need for the present study if this evidence is already known? The reference cited seems to be about active case finding rather than community adherence groups for TPT.

Methods

2. Line 132 - The description reports purposive sampling. What criteria were used? How representative were the study participants of the stakeholder population?

3. Lines 140 – 145 describe three components of a differentiated service delivery model. It is unclear if the study participants experienced a service delivery model that included these components, or if they were presented as a hypothetical delivery model. Additionally, are the three components intended to represent alternative delivery models or parts of the same delivery model? How do parts 1 and 3 fit in the same model?

Results

4. Line 241 – The finding is that TPT delivery is dependent on a constant drug supply. How is this specific to the care model being evaluated since clinic-based delivery would also require that drugs are available?

Discussion

5. Lines 309 – 312 – Stating that the participants perceived that a delivery model was beneficial makes it sound like they experienced this model and contributes to confusion about whether this model is hypothetical or has been implemented.

Minor

6. CAG and community adherence group (spelled out) seem to be used at various points in the manuscript. I suggest using CAG after initial definition unless it is needed for clarity.

Abstract

7. Line 76 – The sentence ends during a phrase.

Introduction

8. Line 94 – “Barriers to TPT…” should cite the source of the information.

Methods

9. Line 143 – define HH

Results

10. Line 200 – “One of the key…” is unclear if it refers to a finding from the current study or prior research.

11. Line 208 – What evidence supports the finding that community based TPT reduces provider workload at the health facility? This seems like an important finding from providers, but the only evidence cited is a finding from caregivers and not providers.

Discussion

12. Line 332 – The sentence is incomplete.

13. Line 339 – The sentence is incomplete.

**Do you want your identity to be public for this peer review?** For information about this choice, including consent withdrawal, please see our Privacy Policy..

Reviewer #1: **Yes:** Christiaan MulderChristiaan MulderChristiaan MulderChristiaan Mulder

Reviewer #2: No

---

## [Decision Letter · Decision Letter 1]

16 Mar 2026

Perceived benefits of community-based TB preventive treatment in children in Uganda: “When she sees other children getting the same medication, she will feel not alone.”

PGPH-D-25-02798R1

Dear Dr. Kakande,

We are pleased to inform you that your manuscript 'Perceived benefits of community-based TB preventive treatment in children in Uganda: “When she sees other children getting the same medication, she will feel not alone.”' has been provisionally accepted for publication in PLOS Global Public Health.

Best regards,

Abraham D. Flaxman, Ph.D.

Academic Editor

Reviewer Comments (if any, and for reference):

Reviewer's Responses to Questions

**Comments to the Author**

Reviewer #3: All comments have been addressed

publication criteria? Is the manuscript technically sound, and do the data support the conclusions? The manuscript must describe methodologically and ethically rigorous research with conclusions that are appropriately drawn based on the data presented.? Is the manuscript technically sound, and do the data support the conclusions? The manuscript must describe methodologically and ethically rigorous research with conclusions that are appropriately drawn based on the data presented.

Reviewer #3: Yes

3. Has the statistical analysis been performed appropriately and rigorously?

Reviewer #3: N/A

4. Have the authors made all data underlying the findings in their manuscript fully available (please refer to the Data Availability Statement at the start of the manuscript PDF file)?

The PLOS Data policy requires authors to make all data underlying the findings described in their manuscript fully available without restriction, with rare exception. The data should be provided as part of the manuscript or its supporting information, or deposited to a public repository. For example, in addition to summary statistics, the data points behind means, medians and variance measures should be available. If there are restrictions on publicly sharing data—e.g. participant privacy or use of data from a third party—those must be specified.requires authors to make all data underlying the findings described in their manuscript fully available without restriction, with rare exception. The data should be provided as part of the manuscript or its supporting information, or deposited to a public repository. For example, in addition to summary statistics, the data points behind means, medians and variance measures should be available. If there are restrictions on publicly sharing data—e.g. participant privacy or use of data from a third party—those must be specified.

Reviewer #3: Yes

5. Is the manuscript presented in an intelligible fashion and written in standard English?

Reviewer #3: Yes

Reviewer #3: Thank you for this revision and for addressing the reviewer comments. Unfortunately, the original reviewers were not available to read it again. I think they will be satisfied with your revisions when they see them in print, however.

I found one additional minor typo during my review:

Line 422 (in track-changes version): “community CAG” is redundant, just say “CAG”

**Do you want your identity to be public for this peer review?** For information about this choice, including consent withdrawal, please see our Privacy Policy..

Reviewer #3: **Yes:** Abraham FlaxmanAbraham FlaxmanAbraham FlaxmanAbraham Flaxman
